# Inhibitory Effect of Adsorption of *Streptococcus mutans* onto Scallop-Derived Hydroxyapatite

**DOI:** 10.3390/ijms241411371

**Published:** 2023-07-12

**Authors:** Momoko Usuda, Mariko Kametani, Masakazu Hamada, Yuto Suehiro, Saaya Matayoshi, Rena Okawa, Shuhei Naka, Michiyo Matsumoto-Nakano, Tatsuya Akitomo, Chieko Mitsuhata, Kazuya Koumoto, Keiko Kawauchi, Takahito Nishikata, Masatoshi Yagi, Toshiro Mizoguchi, Koki Fujikawa, Taizo Taniguchi, Kazuhiko Nakano, Ryota Nomura

**Affiliations:** 1Department of Pediatric Dentistry, Graduate School of Biomedical and Health Sciences, Hiroshima University, Hiroshima 734-8553, Japan; musuda@hiroshima-u.ac.jp (M.U.); mrysk25@hiroshima-u.ac.jp (M.K.); takitomo@hiroshima-u.ac.jp (T.A.); chiekom@hiroshima-u.ac.jp (C.M.); 2Department of Oral & Maxillofacial Oncology and Surgery, Osaka University Graduate School of Dentistry, Suita 565-0871, Japan; hamada.masakazu.dent@osaka-u.ac.jp; 3Department of Pediatric Dentistry, Osaka University Graduate School of Dentistry, Suita 565-0871, Japan; suehiro.yuto.dent@osaka-u.ac.jp (Y.S.); matayoshi.saaya.dent@osaka-u.ac.jp (S.M.); okawa.rena.dent@osaka-u.ac.jp (R.O.); nakano.kazuhiko.dent@osaka-u.ac.jp (K.N.); 4Joint Research Laboratory of Next-Generation Science for Oral Infection Control, Osaka University Graduate School of Dentistry, Suita 565-0871, Japan; yagi@pharmacrea.com (M.Y.); t_mizoguchi@takeda-co.com (T.M.); k_fujikawa@takeda-co.com (K.F.); taniguchi@pharmacrea.com (T.T.); 5Department of Pediatric Dentistry, Okayama University Graduate School of Medicine, Dentistry and Pharmaceutical Sciences, Okayama 700-8558, Japan; nshuhei@okayama-u.ac.jp (S.N.); mnakano@cc.okayama-u.ac.jp (M.M.-N.); 6Faculty of Frontiers of Innovative Research in Science and Technology (FIRST), Konan University, Kobe 650-0047, Japan; koumoto@konan-u.ac.jp (K.K.); kawauchi@konan-u.ac.jp (K.K.); nisikata@konan-u.ac.jp (T.N.); 7Pharmacrea Kobe Co., Ltd., Kobe 651-0085, Japan; 8TSET Co., Ltd., Kariya 448-0022, Japan

**Keywords:** scallop-derived hydroxyapatite, *Streptococcus mutans*, adsorption, RNA sequence, bacterial growth

## Abstract

Hydroxyapatite adsorbs various substances, but little is known about the effects on oral bacteria of adsorption onto hydroxyapatite derived from scallop shells. In the present study, we analyzed the effects of adsorption of *Streptococcus mutans* onto scallop-derived hydroxyapatite. When scallop-derived hydroxyapatite was mixed with *S. mutans*, a high proportion of the bacterial cells adsorbed onto the hydroxyapatite in a time-dependent manner. An RNA sequencing analysis of *S. mutans* adsorbed onto hydroxyapatite showed that the upregulation of genes resulted in abnormalities in pathways involved in glycogen and histidine metabolism and biosynthesis compared with cells in the absence of hydroxyapatite. *S. mutans* adsorbed onto hydroxyapatite was not killed, but the growth of the bacteria was inhibited. Electron microscopy showed morphological changes in *S. mutans* cells adsorbed onto hydroxyapatite. Our results suggest that hydroxyapatite derived from scallop shells showed a high adsorption ability for *S. mutans*. This hydroxyapatite also caused changes in gene expression related to the metabolic and biosynthetic processes, including the glycogen and histidine of *S. mutans*, which may result in a morphological change in the surface layer and the inhibition of the growth of the bacteria.

## 1. Introduction

Dental caries is considered an endogenous infection caused by commensal oral microbiota [1]. When homeostasis in biofilms becomes disrupted during acidification, streptococci, the most widely distributed species in the oral cavity, can adapt to and survive in the acidic environment [2]. The acid-adapted bacteria gain a selective advantage over other species, resulting in the development of dental caries lesions [1]. *S. mutans* can increase in abundance in the acidic environment and promote the progression of dental caries through the loss of minerals on the tooth surface [3].

Antibiotics are used for the prevention of dental caries because of their bactericidal effects and inhibitory effects on biofilm formation [4]. However, antibiotics have drawbacks, such as the emergence of resistant bacteria and drug allergies [5]. Hydroxyapatite occurs in a variety of natural resources, including bones or teeth of vertebrates and minerals, and is a safe material with a good biocompatibility [6]. Hydroxyapatite is a potential adsorbent for the removal of various substances, including bacteria [7,8].

Hydroxyapatite adsorbs on commensal bacteria associated with systemic diseases, such as *Staphylococcus aureus*, *Escherichia coli*, *Enterococcus faecalis*, and oral bacteria, such as *Aggregatibacter actinomycetemcomitans*, *Fusobacterium nucleatum*, *Candida albicans*, and *S. mutans* [9,10]. Hydroxyapatite is also known to have antibacterial properties, and hydroxyapatite uses ions to exert its antibacterial effect, but the detailed mechanism is unknown [9]. Modified hydroxyapatite with a more effective antimicrobial effect has been synthesized by structurally causing ionic substitution in hydroxyapatite [9].

The importance of plaque control for maintaining oral health has led to a continuous search for innovative oral hygiene products, one of which is toothpaste containing hydroxyapatite [11,12]. In the dental field, hydroxyapatite is often focused on enamel remineralization rather than its effect on oral bacteria. However, recent in vitro studies have shown that toothpaste containing hydroxyapatite reduces saliva’s oral bacterial counts and biofilm levels [13,14].

Hydroxyapatite is derived from biogenic sources including animal bones, seashells, and eggshells, which have a high biocompatibility [15]. Hydroxyapatite derived from seashells and eggshells is considered at less of a risk of immunological rejection and infection than other hydroxyapatite [16]. Seashell-derived hydroxyapatite has been utilized in developing and clinically applying biomedical materials [17]. Seashell-derived hydroxyapatite is a bone substitute in bone reconstruction surgery, has a low toxicity to osteoblast precursor cells, and can effectively induce cell differentiation [18]. Hydroxyapatite derived from seashells can be applied to novel coatings for metallic dental implants due to its good cytocompatibility and antibacterial activity [10].

Research on hydroxyapatite’s effects on bacteria can focus on its properties, its impact on living organisms, and the changes that bacteria undergo. There is a steadily increasing number of reports on the properties of hydroxyapatite concerning the substitution of various ions [19]. In addition, research for bio-applications of hydroxyapatite in the fields of bone tissue engineering, remineralization of teeth, dental implants, and pharmacy has been progressing [20,21]. In bacteriology, hydroxyapatite or substituted ions inhibit bacterial activity by adsorbing various bacteria [9]. However, studies focusing on the details of the genetic level of the bacteria in the presence of hydroxyapatite are incredibly scarce. Therefore, in the present study, we decided to use bioinformatics to clarify the changes in gene expression of *S. mutans* in the presence of hydroxyapatite.

## 2. Results

### 2.1. Adsorption of S. mutans onto Scallop-Derived Hydroxyapatite

Scallop-derived hydroxyapatite powder of various concentrations (0%, 0.01%, 0.1%, 1%, and 10%) was added to 1.0 × 10^9^ colony-forming units (CFU)/mL of *S. mutans* MT8148 suspended in phosphate-buffered saline (PBS), and the bacteria were reacted with the hydroxyapatite by vortexing for 10 s. The number of *S. mutans* adsorbed onto the hydroxyapatite was determined. Immediately after the vortexing, the 0.01% and 0.1% hydroxyapatite-added groups hardly adsorbed *S. mutans* (Figure 1). In contrast, 8.5 × 10^8^ CFU/mL and 1.0 × 10^9^ CFU/mL of *S. mutans* were adsorbed onto hydroxyapatite in the 1% and 10% hydroxyapatite-added groups, respectively. Over time, the adsorption of *S. mutans* progressed even in the groups with lower concentrations of hydroxyapatite. After 24 h, the number of *S. mutans* in PBS without added hydroxyapatite was 6.8 × 10^8^ CFU/mL, and the number of *S. mutans* adsorbed onto hydroxyapatite ranged from 5.3 × 10^8^ CFU/mL to 6.8 × 10^8^ CFU/mL.

### 2.2. RNA Sequencing Analysis of S. mutans Treated with Scallop-Derived Hydroxyapatite

Using the method described above, *S. mutans* reacted with scallop-derived hydroxyapatite (0%, 0.1%, 1%, and 10%) for 10 s, and then stood at 37 °C for 24 h. RNA sequencing was performed to comprehensively analyze the gene expression changes that adsorption onto the hydroxyapatite causes in *S. mutans*. Figure 2 shows the schedule of this experiment as a schema. First, 1398 filter-passing genes were found from among 2042 detected genes. Then, we compared the data for different concentrations of hydroxyapatite: 0% vs. 0.1%, 0% vs. 1.0%, and 0% vs. 10%. In each condition, the top 5% upregulated and downregulated genes were identified. Among these, 6 upregulated genes and 15 downregulated genes were common to all three conditions. There were 156 upregulated genes and 142 downregulated genes detected in at least one of the three conditions. These data were used for a bioinformatic analysis, including protein–protein interaction (PPI) network and Gene Ontology (GO) enrichment analyses.

### 2.3. Protein–Protein Interaction Network Analysis of Upregulated and Downregulated Genes of S. mutans Treated with Scallop-Derived Hydroxyapatite

The PPI network analysis was performed on the 6 upregulated and 15 downregulated genes found in all three sets of conditions. Among the upregulated genes, 5 of the 6 (*citG2*, *glgD*, *trk*, SMU_311, and SMU_1487, but not SMU_1230c) were in a network of ≥5 genes, with the greatest interaction around *glgD* (Figure 3a). In contrast, only 3 of the 15 downregulated genes (*phnA*, SMU_10, and SMU_112c) were in a network of ≥5 genes (Figure 3b).

### 2.4. Gene Ontology Enrichment Analysis of Genes of S. mutans Upregulated on Treatment with Scallop-Derived Hydroxyapatite

The GO enrichment analysis was performed using ShinyGO for the 156 upregulated and 142 downregulated genes of *S. mutans* in the presence of hydroxyapatite (i.e., genes that were among the top 5% of up/downregulated genes in at least one of the 0% vs. 0.1%, 0% vs. 1%, and 0% vs. 10% hydroxyapatite conditions). The 156 upregulated genes were found to be related to pathways including the metabolism and biosynthesis of glycogen and histidine (Figure 4a,b). Interactive plots that show the relationships between enriched pathways highlighted interactions involving carbohydrate metabolism, including glycogen metabolism and biosynthetic processes; amino acid metabolism, including histidine metabolism and biosynthetic processes; and the phosphotransferase system (Figure 4c). In contrast, the analysis of the 142 downregulated genes had no pathway.

### 2.5. Principal Component Analysis and Heat Map Constructed Using Upregulated and Downregulated Genes of S. mutans Treated with Scallop-Derived Hydroxyapatite

A principal component analysis (PCA) using the 1398 filter-passing genes showed that PC1 (which accounted for 51.8% of the variance) had a peak change at 0.1% hydroxyapatite, with smaller changes at 1% and 10% (Figure 5a). PC2 (29.6%) showed a hydroxyapatite-concentration-dependent increase in the change. A heatmap representing the expression level of the 1398 filter-passing genes showed an accumulation of 109 upregulated genes in the presence of 10% hydroxyapatite (Figure 5b,c).

### 2.6. GO Enrichment Analysis of 109 Genes of S. mutans Upregulated in the Presence of 10% Scallop-Derived Hydroxyapatite

The GO enrichment analysis was performed using ShinyGO for the 109 upregulated genes of *S. mutans* that specifically accumulated in the presence of 10% scallop-derived hydroxyapatite (Figure 6a,b). The 109 upregulated genes were related to pathways, including histidine metabolic and biosynthetic processes, the aromatic amino acid family metabolic process, and the phosphotransferase system. Interactive plots show relationships between enriched pathways involving amino acid metabolism, including histidine metabolism and biosynthesis (Figure 6c).

### 2.7. Analysis of Bacterial Growth and Morphology of S. mutans in the Presence of Scallop-Derived Hydroxyapatite

Adsorption of *S. mutans* onto hydroxyapatite caused changes in gene expression that indicated abnormalities in metabolic and biosynthetic processes. We hypothesized that the presence of hydroxyapatite might kill *S. mutans*; however, the number of *S. mutans* adsorbed on hydroxyapatite was not significantly different from that of *S. mutans* in PBS for the same time period (Appendix A).

Next, we analyzed the bacterial growth and the morphology of *S. mutans* adsorbed on hydroxyapatite because abnormalities in metabolic or biosynthetic processes of carbohydrate and histidine can cause growth inhibition and morphological changes of bacteria [22,23,24]. When 1.0 × 10^7^ CFU/mL of *S. mutans* was reacted with hydroxyapatite for 10 s and was grown in a Brain Heart Infusion (BHI) broth at 37 °C, 0.01% and 0.1% hydroxyapatite had little effect on the bacterial growth (Figure 7a). The 1% hydroxyapatite reduced the number of *S. mutans* up to 6 h, but the bacteria grew after 6 h. Ten percent hydroxyapatite reduced the number of *S. mutans* bacteria at all time points. Scanning electron microscopy of *S. mutans* adsorbed onto hydroxyapatite in PBS showed that the outline of the bacteria was poorly marginated (Figure 7b).

## 3. Discussion

In the present in vitro study, we focused on treatment with hydroxyapatite derived from scallop shells as a novel method to inhibit *S. mutans*. The use of hydroxyapatite derived from scallop shells, which was originally an industrial waste product [25,26], is meaningful in terms of environmental protection. The scallop shells contain magnesium and other marine minerals, and they have a better biocompatibility than the widely known mineral-derived hydroxyapatite [27].

Many kinds of toothpaste containing hydroxyapatite have been developed [28], but most relevant studies have focused on how and whether they strengthen the dental structure, rather than the adsorption of oral bacteria as their effect. Our results show that hydroxyapatite derived from scallop shells has a high ability to adsorb *S. mutans*. At a high concentration (1% to 10%), this hydroxyapatite adsorbed *S. mutans* within a few minutes, and at a low concentration (0.01%, 0.1%), it showed an adsorption effect within a few hours. These results suggest that the application of scallop-derived hydroxyapatite in daily oral hygiene may be effective in lessening *S. mutans* in the oral cavity.

Hydroxyapatite adsorbs various bacteria [29]. However, no studies have analyzed the adsorption of *S. mutans* onto shell-derived hydroxyapatite and its effect on gene expression in *S. mutans*. Here, we identified changes in gene expression in *S. mutans* induced by adsorption onto scallop-derived hydroxyapatite. RNA sequencing results showed that multiple genes of *S. mutans* were up- or downregulated after adsorption onto hydroxyapatite. The GO enrichment analysis using these genes revealed pathways in which the upregulated genes were involved that may have significant physiological effects on *S. mutans*.

A PPI network analysis and GO enrichment analysis are widely used following gene expression analyses for clustering potentially up- or downregulated proteins and clarifying the biological significance of the obtained clusters, respectively [30,31]. Nevertheless, few studies have reported on the use of these methods for *S. mutans* [32,33]. In the present study, the PPI network analysis using genes whose expression increased in the presence of hydroxyapatite revealed that the *glg* operon, including *glgA*, *glgB*, *glgC*, and *glgD*, was part of the largest network. The *glg* operon is involved in the synthesis of glycogen via ADP–glucose from glucose-1-phosphate produced from glucose upstream of the Emden–Meyerhof pathway [34,35]. Therefore, *S. mutans* adsorbed onto hydroxyapatite may store excess glycogen. Glycogen metabolism and biosynthesis processes were also identified in the GO enrichment analysis and this may be the major effect of scallop-derived hydroxyapatite on *S. mutans*.

PCA is widely used as a method to overview gene expression changes [36]. Here, PC1 in the PCA showed a peak of change at a low concentration (0.1%) of hydroxyapatite, indicating a major difference in gene expression in *S. mutans* in the presence and absence of hydroxyapatite. PC2 showed a hydroxyapatite-concentration-dependent increase in the gene expression change. A heatmap indicated that 109 upregulated genes accumulated in the presence of a high concentration (10%) of hydroxyapatite compared with the absence of the hydroxyapatite. We performed the GO enrichment analysis using these 109 upregulated genes, which mainly highlighted an effect on histidine metabolic and biosynthetic processes; this result indicates that a high concentration of hydroxyapatite particularly affects histidine metabolism and biosynthesis.

Hydroxyapatite inhibited the growth of *S. mutans* and caused changes in the morphology of the bacteria. Studies have reported that in some bacteria, the abnormal expression of genes relating to carbohydrate metabolism results in the inhibition of bacterial growth and changes in the cell shape [22,23,37,38]. In addition, *S. mutans* showed growth inhibition and morphological changes when genes involved in histidine phosphorylation were mutated [24]. Furthermore, links between the inhibition of bacterial growth and cell shape have been reported [39,40]. Therefore, *S. mutans* adsorbed onto hydroxyapatite may undergo reciprocal processes of metabolic overexpression, growth inhibition, and morphological change. The oral microbiome has been the focus of much research in recent years [41]; a detailed analysis of the effects of hydroxyapatite on this microbiome is needed in the future.

Our study has certain limitations: the first concerns hydroxyapatite’s zeta potential. The zeta potential of ionic non-substituted hydroxyapatite is a negative charge [42,43], and the zeta potential of hydroxyapatite used in this study is unlikely to favor adhesion to *S. mutans* since the zeta potential of *S. mutans* is also a negative charge. According to recent reports, ion doping has changed the zeta potential of hydroxyapatite to a positive charge and acquired higher antibacterial properties [44,45]. Therefore, it is necessary to prepare hydroxyapatite with a positive zeta potential and analyze the changes in the gene expression of *S. mutans* in the presence of hydroxyapatite to achieve a higher inhibitory effect against *S. mutans*. The second limitation is regarding the ions of hydroxyapatite. Since ions released from antimicrobial substances can alter the adhesion capacity and gene expression of bacteria [46], ions released from hydroxyapatite can alter the gene expression of *S. mutans*. In this study, we used non-substituted hydroxyapatite with ions derived from shells. In the future, hydroxyapatite derived from various biogenic sources and hydroxyapatite substituted with ions should be analyzed for the relationship between the amount of ionic release and changes in the gene expression of *S. mutans*.

## 4. Materials and Methods

### 4.1. S. mutans Strain and Culture Conditions

The *S. mutans* strain MT8148 (serotype *c*) was cultured on Mitis Salivarius agar plates (Difco Laboratories) containing bacitracin (0.2 U/mL; Sigma-Aldrich, St. Louis, MO, USA) and 15% (*w*/*v*) sucrose (MSB-agar) at 37 °C for 48 h [47]. A single colony was inoculated into the BHI broth and cultured at 37 °C for 18 h and used in subsequent studies.

### 4.2. Scallop-Derived Hydroxyapatite

This study used a conventional wet method to prepare the hydroxyapatite derived from scallop shells [48]. The hydroxyapatite was ionically non-substituted and confirmed using an X-ray Fluorescence Analysis (XRF), Energy Dispersive X-ray Spectroscopy (EDX), and ion chromatography.

### 4.3. The Adsorption of S. mutans onto Scallop-Derived Hydroxyapatite

Hydroxyapatite powder was obtained from scallop shells collected in Hokkaido, Japan, and was provided by TSET Co. (Aichi, Japan). Cultured bacteria were collected using centrifugation at 1000× *g* at 4 °C for 10 min. The cultures were washed and resuspended in PBS to an OD_550_ value of 1.0, which corresponds to 1 × 10^9^ CFU/mL. Scallop-derived hydroxyapatite powder was then added to the bacterial suspensions at final concentrations of 0%, 0.01%, 0.1%, 1%, and 10%. The bacterial suspension was vortexed for 10 s to allow the bacteria to react with hydroxyapatite, and then stood at 37 °C for 5 min, 30 min, 3 h, or 24 h. Then, 100 µL of liquid containing *S. mutans* that had not adsorbed on the hydroxyapatite and precipitated at the bottom of the solution was collected from the top of the vessel. This potential bacterial suspension was cultured on MSB-agar at 37 °C for 48 h. The number of *S. mutans* adsorbed onto hydroxyapatite was calculated by subtracting the number of *S. mutans* not adsorbed on hydroxyapatite in the hydroxyapatite-added group at each time point from the number of *S. mutans* in the group without hydroxyapatite at each time point. In addition, the bacterial suspension of *S. mutans* that reacted with hydroxyapatite was sonicated and cultured on MSB-agar at 37 °C for 48 h to determine the total number of bacteria. All assays were carried out three times, and mean and standard deviation values were determined.

### 4.4. RNA Sequencing and FASTQ File Processing

*S. mutans* was treated with scallop-derived hydroxyapatite at various concentrations (0.1%, 1%, or 10%) for 24 h. Bacterial cells were lysed using Qiazol (Qiagen, Germantown, MD, USA) and the total RNA of *S. mutans* was isolated using an miRNeasy Micro Kit (Qiagen) according to the manufacturer’s instructions. Library preparation was performed using a GenNext RamDA-seq Single Cell Kit (Toyobo, Tokyo, Japan). Whole transcriptome sequencing was executed with an Illumina NovaSeq 6000 platform in the 100-base single-end mode. Sequenced reads were mapped to the reference genome sequence (*S. mutans* UA159; GenBank Accession: NC_004350.2) using HISAT2 ver. 2.1.0. Counts per gene were calculated with featureCounts v2.0.0.

### 4.5. Analysis of RNA Sequencing Count Data

The RNA sequencing counts were imported into Subio Platform v1.24.5853 (Subio Inc.; https://www.subioplatform.com, accessed on 8 March 2023) for the preprocessing, filtering, and extraction of differentially expressed genes. In the preprocessing, we turned the counts into log_2_ values and applied global normalization at the 90th percentile. Furthermore, we set the lower limit as 20 in the linear scale by replacing counts <20 with 20. Finally, we calculated the log_2_ ratio compared with the control sample. In the filtering, we excluded counts of tRNAs, rRNAs, and genes that were always <20 or had log_2_ ratios in the range −0.25 to 0.25 in all samples; 1398 genes remained after this filtering. We extracted the top 5% of up- or downregulated genes at each hydroxyapatite concentration from these filter-passed genes.

### 4.6. Bioinformatic Analysis

The GO enrichment analysis was performed using ShinyGO 0.77 online resources (http://bioinformatics.sdstate.edu/go/, accessed on 8 March 2023). A *p*-value cut-off of 0.05 for the false discovery rate was used to determine the genes used for the GO enrichment analysis. We used these genes to establish PPI networks based on the StringApp11.5 (Search Tool for the Retrieval of Interacting Genes/Proteins) online database (https://string-db.org/, accessed on 8 March 2023). Then, the most significant modules in the PPI networks were visualized. In addition, we performed PCA using the filtered genes.

### 4.7. Bacterial Growth Assay

A bacterial growth assay was performed in accordance with a previously described method, with some modifications [49,50]. Briefly, cultured bacteria were added to the BHI broth at a final concentration of 1.0 × 10^7^ CFU/mL in the presence of scallop-derived hydroxyapatite powder at a final concentration of 0%, 0.01%, 0.1%, 1%, and 10%. The bacterial suspensions were vortexed for 10 s to adsorb bacteria onto the hydroxyapatite. The bacterial mixtures were cultured at 37 °C for 3, 6, 12, and 24 h and then spread onto MSB-agar plates. The plates were incubated anaerobically at 37 °C for 48 h and the number of colonies was counted. All assays were carried out three times, and mean and standard deviation values were determined.

### 4.8. Electron Microscopy

An observation using electron microscopy was performed in accordance with a previously described method [51,52]. As a preparation for scanning electron microscopy (SEM) imaging, each bacterial sample was washed and fixed with 2% osmium tetroxide and 1% glutaraldehyde, dehydrated with ethanol, and then dried with *t*-butyl alcohol with the freeze-drying method. The dried samples were mounted on the stage and coated with osmium for conductive processing and then observed with SEM.

### 4.9. Statistical Analysis

GraphPad Prism 9 software (GraphPad Software Inc., La Jolla, CA, USA) was used for statistical analyses. Comparisons between two groups were performed using a Student’s *t*-test. Differences between multiple groups for each assay were determined using an analysis of variance. Bonferroni correction was used for a post hoc analysis. Results were considered significantly different at *p* < 0.05.

## 5. Conclusions

Hydroxyapatite derived from scallop shells showed a high adsorption capacity for *S. mutans*. A comprehensive genetic analysis using RNA sequencing revealed that in hydroxyapatite, changes in metabolic and biosynthetic processes, especially those related to glycogen and histidine, of *S. mutans* were promoted, as well as the inhibition of the bacterial growth and morphological changes. Based on the results of our in vitro study, this hydroxyapatite may influence the gene network of *S. mutans* and help maintain oral health by decreasing the risk of dental caries induction.

## Figures and Tables

**Figure 1 ijms-24-11371-f001:**
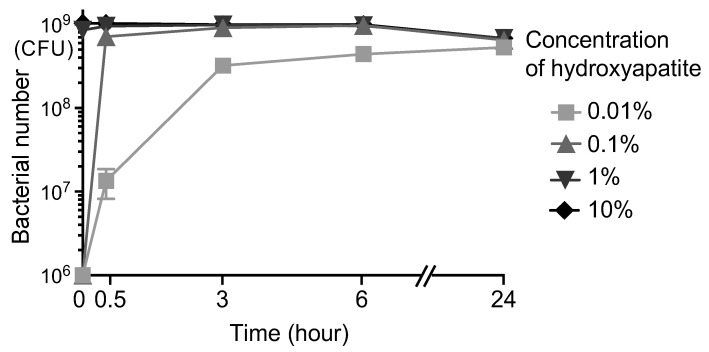
Adsorption of *S. mutans* onto scallop-derived hydroxyapatite. The number of *S. mutans* adsorbed onto scallop-derived hydroxyapatite after the reaction of the bacteria with the hydroxyapatite by vortexing for 10 s.

**Figure 2 ijms-24-11371-f002:**
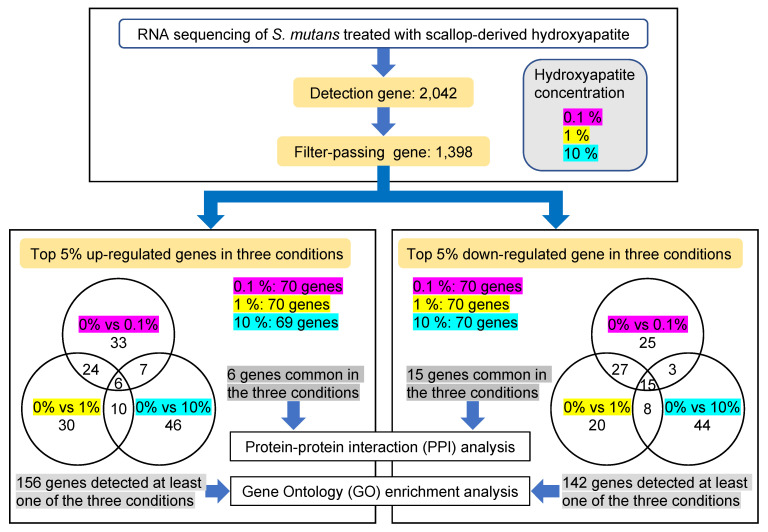
Schema of analyses using data from RNA sequencing of *S. mutans* treated with scallop-derived hydroxyapatite.

**Figure 3 ijms-24-11371-f003:**
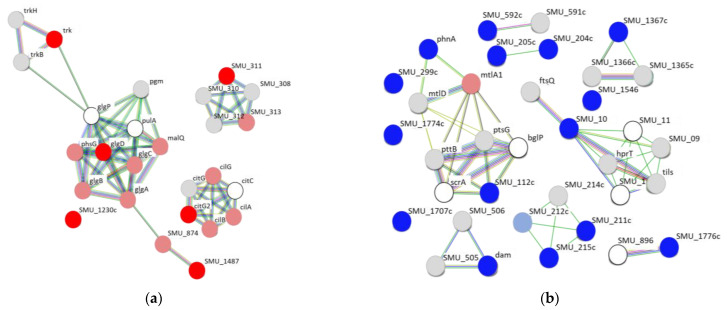
Protein–protein interaction (PPI) network analysis of (**a**) 6 upregulated and (**b**) 15 downregulated genes observed among the top 5% of up- or downregulated genes when *S. mutans* was treated with 0.1%, 1%, and 10% scallop-derived hydroxyapatite. The genes upregulated at all the concentrations of hydroxyapatite are shown in dark red, and the genes downregulated at all the concentrations are shown in dark blue. Genes upregulated at any of the concentrations are shown in pale red, and genes downregulated at any of the concentrations are shown in pale blue. Gray indicates genes whose expression did not change significantly in RNA sequencing of *S. mutans* treated with scallop-derived hydroxyapatite. White indicates that a gene was not detected.

**Figure 4 ijms-24-11371-f004:**
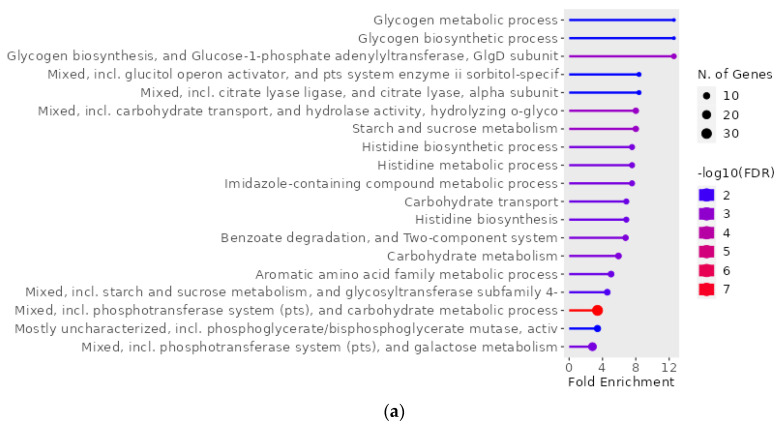
Gene Ontology (GO) enrichment analysis of 156 genes upregulated in *S. mutans* treated with scallop-derived hydroxyapatite (genes that were among the top 5% of upregulated genes in at least one of the 0% vs. 0.1%, 0% vs. 1%, and 0% vs. 10% hydroxyapatite conditions). (**a**) GO enrichment analysis was performed with ShinyGO and the pathways found are shown. (**b**) Hierarchical clustering tree summarizing correlations between pathways. Pathways with many shared genes are clustered together. Larger dots indicate more significant *p*-values. (**c**) Interactive plot showing the relationships between enriched pathways. Two pathways (nodes) are connected when they share ≥20% of genes. Darker nodes are more significantly enriched gene sets, and larger nodes represent larger gene sets. Thicker edges indicate more overlapping genes.

**Figure 5 ijms-24-11371-f005:**
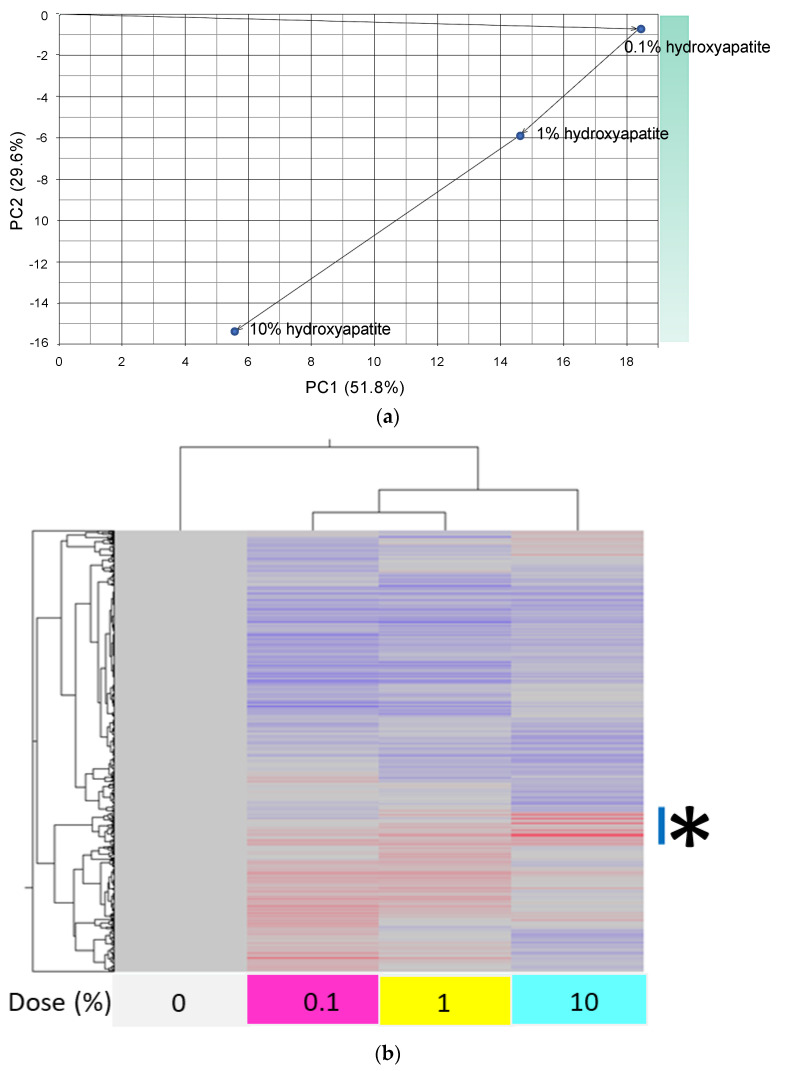
Principal component analysis (PCA) and heat map using 1398 genes up- or downregulated when *S. mutans* was treated with scallop-derived hydroxyapatite. (**a**) PCA of *S. mutans* in the presence of 0.1%, 1%, and 10% hydroxyapatite. (**b**) Heatmap and clustering of 1398 filter-passing upregulated or downregulated genes. (**c**) Detailed heatmap and clustering of 109 genes extracted from the part of (**b**) marked *.

**Figure 6 ijms-24-11371-f006:**
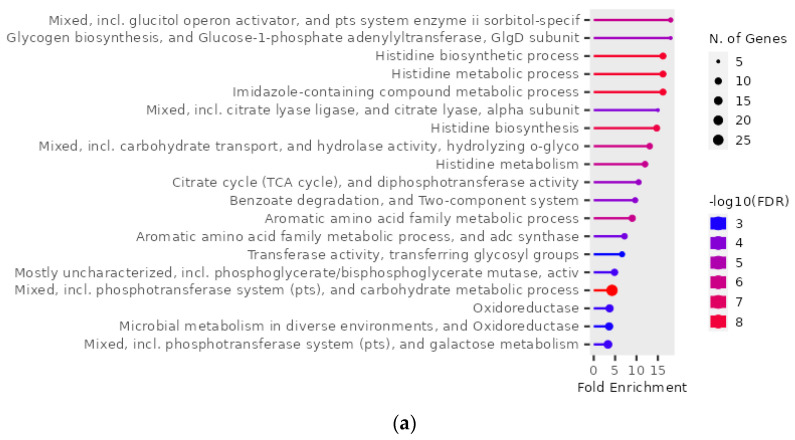
GO enrichment analysis of 109 genes of *S. mutans* that are specifically upregulated in the presence of 10% scallop-derived hydroxyapatite compared with the absence of the hydroxyapatite. (**a**) GO enrichment analysis was performed on the 109 upregulated genes indicated in Figure 5c with ShinyGO and the pathways found are shown. (**b**) Hierarchical clustering tree summarizing correlations between pathways listed in the enrichment table pathways with many shared genes being clustered together. Larger dots indicate more significant *p*-values. (**c**) Interactive plot showing the relationships between enriched pathways. Two pathways (nodes) are connected when they share 20% or more of genes. Darker nodes are more significantly enriched gene sets, and larger nodes represent larger gene sets. Thicker edges indicate more overlapping genes.

**Figure 7 ijms-24-11371-f007:**
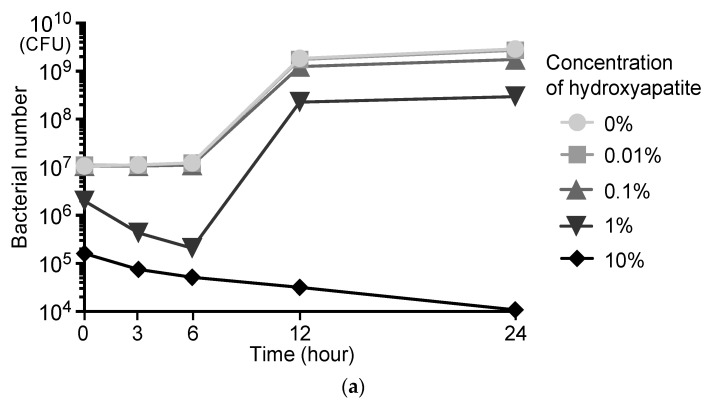
Changes in bacterial growth and morphology of *S. mutans* adsorbed onto scallop-derived hydroxyapatite. (**a**) Growth of *S. mutans* treated with scallop-derived hydroxyapatite. (**b**) Representative scanning electron microscopy images of *S. mutans* adsorbed on scallop-derived hydroxyapatite. The lower panels show high-magnification images of the boxed regions in the upper images. White arrowheads indicate bacteria. Bars = 2 μm (upper images) and 500 nm (lower images).

## Data Availability

The data are available from the corresponding author upon reasonable request.

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
