# Peer review of "Inhibitory Effect of Adsorption of Streptococcus mutans onto Scallop-Derived Hydroxyapatite"

_ijms, 2023, doi:10.3390/ijms241411371_

Round 1

Reviewer 1 Report

This manuscript explores an intriguing and novel topic, where the authors have made commendable efforts. However, there are certain sections that require careful improvement.

- The English language usage needs further revision before reaching to the final decision.

- The aim of the abstract is unclear due to the expression "…of adsorption onto…" in English. Please rephrase it (lines 30-31).

- The introduction is concise and would benefit from expansion to provide more context. Consider incorporating relevant in vitro studies on "oral health" and "toothpaste containing hydroxyapatite" for better coherence. Some suggested studies are: [Int J Environ Res Public Health. 2022;19(13):8056. doi: 10.3390/ijerph19138056.]; [Appl. Sci. 2020, 10, 6721. https://doi.org/10.3390/app10196721].

- The aim of the present study is not clearly understandable and needs to be emphasized more effectively (lines 62-65).

- In Figure 1; 7 (a); S1, the graphs need to be recreated with improved axis labels. Specifically, one of the axes should represent time to establish a clear correlation. The legends alone are insufficient for understanding the timing of each graph.

- In line 203, the authors need to provide better clarification regarding the type of study, such as specifying whether it is an in vitroin situ, or in vivo study.

- The first sentence of the conclusions section is unclear in its current form (line 335). Please rephrase it to improve clarity. Additionally, consider enhancing the entire section for better coherence and effectiveness.

The English needs to be improved in several sentences starting from the abstract.

Reviewer 2 Report

The submitted manuscript entitled “Inhibitory effect of adsorption of Streptococcus mutans onto scallop-derived hydroxyapatite” is very interesting, however, it needs improvements before publishing. Below authors can find suggestions and comments that can help in improving the manuscript. I hope the authors will find them useful.

The abstract is very well written.

The introduction is too short. The authors should explain what was previously done on hydroxyapatite derived from seashells and bacteria adhesion on the hydroxyapatite in general. The introduction needs to be significantly improved.

Authors should explain the novelty of the study as there are several studies focused on the antibacterial properties of hydroxyapatite substituted or non-substituted with ions and bacteria adhesion on the surface.

How was hydroxyapatite prepared from seashells (calcium carbonate)? Which method was used? Why authors did not characterize obtained hydroxyapatite? What are the trace elements in obtained hydroxyapatite obtained from biogenic sources? Do the ions substitute in hydroxyapatite (due to using a biogenic precursor) affect bacteria adhesion and observed changes in genes expresion?

The Zeta potential of hydroxyapatite is crucial for bacteria adhesion as shown in recent publications on hydroxyapatite and bacterias (the year 2022 and 2023). Does obtained hydroxyapatite have positive or negative zeta potential? This should be correlated with the potential of the bacteria surface, which is negative. Authors should find studies where the zeta potential of hydroxyapatite and bacteria are discussed and reported.

Figures 2,3 4 and 6 should be presented more clearly and the quality of the presentation should be increased.

Why gene expression was changed. Is it related to the released ions from the hydroxyapatite?

Materials and methods should be written in more detail.

Before considering this manuscript for publishing, the material characterization needs to be added.

The conclusion should be improved.
